# Exphormer: Scaling Graph Transformers with Expander Graphs

## Abstract

Graph transformers have emerged as a promising architecture for a variety of graph learning and representation tasks. Despite their successes, though, it remains challenging to scale graph transformers to large graphs while maintaining accuracy competitive with message-passing networks. In this paper, we introduce *Exphormer*, a framework for building powerful and scalable graph transformers. Exphormer consists of a sparse attention mechanism based on *expander graphs*, whose mathematical characteristics, such as spectral expansion, pseduorandomness, and sparsity, yield graph transformers with complexity only linear in the size of the graph, while allowing us to prove desirable theoretical properties of the resulting transformer models. We show that incorporating Exphormer into the recently-proposed GraphGPS framework produces models with competitive empirical results on a wide variety of graph datasets, including state-of-the-art results on three datasets. We also show that Exphormer can scale to datasets on larger graphs than shown in previous graph transformer architectures.

## 1 Introduction

Graph learning has become an important and popular area of study that has yielded impressive results on a wide variety of graphs and tasks, including molecular graphs, social network graphs, knowledge graphs, and more. While much research around graph learning has focused on graph neural networks (GNNs), which are based on local *message-passing*, a more recent approach to graph learning that has garnered much interest involves the use of *graph transformers* (GTs). Graph transformers largely operate by encoding graph structure in the form of a *soft inductive bias*. These can be viewed as a graph adaptation of the Transformer architecture (Vaswani et al., 2017) that are successful in modeling sequential data in applications such as natural language processing.

Graph transformers allow nodes to attend to all other nodes in a graph, allowing for direct modeling of long-range interactions, in contrast to GNNs. This allows them to avoid several limitations associated with local message passing GNNs, such as oversmoothing (Oono & Suzuki, 2020), oversquashing (Alon & Yahav, 2021; Topping et al., 2022), and limited expressivity (Morris et al., 2019; Xu et al., 2018). The promise of graph transformers has led to a large number of different graph transformer models that have been proposed in recent years (Dwivedi & Bresson, 2020; Kreuzer et al., 2021; Ying et al., 2021; Mialon et al., 2021). A major issue with graph transformers is the need to identify the location and structure of nodes within the graph, which has also led to a number of proposals for positional and structural encodings for graph transformers (Lim et al., 2022).

One major challenge for graph transformers is their poor scalability, as the standard global attention mechanism incurs time and memory complexity of $O(|V|^2)$, *quadratic* in the number of nodes in the graph. While this cost is often acceptable for datasets with small graphs (e.g., molecular graphs), it can be prohibitively expensive for datasets containing larger graphs, where graph transformer models often do not fit in memory even for high-memory GPUs, and hence would require much more complex and slower schemes to apply. Moreover, despite the expressivity advantages of graph transformer networks (Kreuzer et al., 2021), graph transformer-based architectures have often lagged message-passing counterparts in accuracy in many practical settings.

A breakthrough came about with the recent advent of GraphGPS (Rampásek et al., 2022), a modular framework for constructing graph transformers by mixing and matching various positional and structural encodings with local message passing and a global attention mechanism. To overcome

the quadratic complexity of the "dense" full transformer, they also incorporate "sparse" attention mechanisms, like Performer (Choromanski et al., 2021) or Big Bird (Zaheer et al., 2020). These sparse trasnformer mechanisms are an attempt at improving the scalability. This combination of Transformers and GNNs achieves state-of-the-art performance on a wide variety of datasets.

Despite great successes, the aforementioned works leave open some important questions. For instance, unlike pure attention-based approaches (e.g., SAN, Graphormer), GraphGPS combines transformers with message passing, which brings into question how much of the realized accuracy gains are due to transformers themeselves. Indeed, Rampásek et al.'s ablation studies showed that the impact of the transformer component of the model is limited: on a number of datasets, higher accuracies can be achieved in the GraphGPS framework by not using attention at all (rather than, say, BigBird). The question remains, then, of whether transformers in their own right can obtain results on par with message-passing based approaches while scaling to large graphs.

Another major question concerns graph transformers' scalability. While BigBird and Performer are linear attention mechanisms, they still incur computational overhead that dominates the per-epoch computation time for moderately-sized graphs. The GraphGPS work tackles datasets with graphs of up to 5,000 nodes, a regime in which the full-attention transformer is in fact computationally faster than many sparse linear-attention mechanisms. Perhaps more suitable sparse attention mechanisms could enable their framework to operate on even larger graphs.

Relatedly, existing sparse attention mechanisms have largely been designed for *sequences*, which are natural for language tasks but behave quite differently from graphs. From the ablation studies, BigBird and Performer are not as effective on graphs, unlike in the sequence world. Thus, it is natural to ask whether one can design sparse attention mechanisms more tailored to learning interactions on general graphs.

**Our contributions.** We design a sparse attention mechanism, EXPHORMER, that has computational cost linear in the number of nodes and edges. We introduce *expander graphs* as a powerful primitive in designing scalable graph transformer architectures. Expander graphs have several desirable properties — small diameter, spectral approximation of a complete graph, good mixing properties — which make them a suitable ingredient in a sparse attention mechanism. As a result, we are able to show that EXPHORMER, which combines expander graphs with global nodes and local neighborhoods, spectrally approximates the full attention mechanism with only a small number of layers, and has universal approximation properties.

Furthermore, we show the efficacy of EXPHORMER within the GraphGPS framework. That is, combining EXPHORMER with GNNs, helps achieve good performance on a number of datasets, including state-of-the-art results on CIFAR10, MNIST, and PATTERN. On many datasets, EXPHORMER is often even *more accurate* than full attention models, indicating that our attention scheme perhaps provides good inductive bias for places the model "should look," while being more efficient and less memory-intensive. Furthermore, EXPHORMER can scale to larger graphs than previously shown — we demonstrate that a pure EXPHORMER model can achieve strong results on ogbn-arxiv, a challenging transductive problem on a citation graph with over 160K nodes and a million edges, a setting in which full transformers are prohibitively expensive due to memory constraints.

## 2 RELATED WORK

**Graph Neural Networks (GNNs).** Early works in the area of graph learning and GNNs include the development of a number of architectures such as GCN (Defferrard et al., 2016; Kipf & Welling, 2017), GraphSage (Hamilton et al., 2017), GIN (Xu et al., 2018), GAT (Veličković et al., 2018), GatedGCN (Bresson & Laurent, 2017), and more. GNNs are based on a message-passing architecture that generally confines their expressivity to the limits of the 1-Weisfeiler-Lehman (1-WL) isomorphism test (Xu et al., 2018).

A number of recent papers have sought to augment GNNs to improve their expressivity. For instance, one approach has been to use *additional features* that allow nodes to be distinguished – such as using a one-hot encoding of the node (Murphy et al., 2019) or a random scalar feature Sato et al. (2021) – or to encode positional or structural information of the graph – e.g., skip-gram based network embeddings (Qiu et al., 2018), substructure counts (Bouritsas et al., 2020), or Laplacian

eigenvectors (Dwivedi et al., 2021). Another direction has been to *modify the message passing rule* to allow the network to take further advantage of the graph structure, including the directional graph networks (DGN) of Beaini et al. (2021) that use Laplacian eigenvectors to define directional flows that are used for anisotropic message aggregation, or – to *modify the underlying graph* over which message passing occurs, *higher-order GNNs* (Morris et al., 2019) or the use of substructures such as junction trees (Fey et al., 2020) and simplicial complexes (Bodnar et al., 2021).

**Graph transformer architectures.**   Attention mechanisms have been extremely successful in sequence modeling since the seminal work of Vaswani et al. (2017). The GAT architecture (Veličković et al., 2018) proposed using an attention mechanism to determine how a node aggregates information from its neighbors. It does not use a positional encoding for nodes, limiting its ability to exploit global structural information. GraphBert (Zhang et al., 2020) uses the graph structure to determine an encoding of the nodes, but not for the underlying attention mechanism.

Graph transformer models typically operate on a fully-connected graph in which every pair of nodes is connected, regardless of the connectivity structure of the original graph. Spectral Attention Networks (SAN) (Kreuzer et al., 2021) make use of *two* attention mechanisms, one on the fully-connected graph and one on the original edges of the input graph, while using Laplacian positional encodings for the nodes. Graphormer (Ying et al., 2021) uses a single dense attention mechanism but adds structural features in the form of centrality and spatial encodings. Meanwhile, GraphiT (Mialon et al., 2021) incorporates relative positional encodings based on diffusion kernels.

GraphGPS (Rampásek et al., 2022) proposed a general framework for combining message-passing networks with attention mechanisms, while allowing for the mixing and matching of positional and structural embeddings. Specifically, the framework also allows for sparse transformer models like BigBird (Zaheer et al., 2020) and Performer (Choromanski et al., 2021).

**Sparse Transformers.**   Standard (dense) transformers have quadratic complexity in the number of tokens, which limits their scalability to extremely long sequences. By contrast, *sparse transformer* models improve computational and memory efficiency by restricting the attention pattern, i.e., the pairs of nodes that can interact with each other. In addition to BigBird and Performer, there have been a number of other proposals for sparse transformers; Tay et al. (2020) provide a survey.

## 3   THE EXPHORMER ATTENTION MECHANISM

This section describes EXPHORMER, our *sparse* generalized attention mechanism that can be used in individual layers of a graph transformer architecture. We begin by describing graph attention mechanisms in general.

### 3.1   ATTENTION MECHANISM ON GRAPHS

An attention mechanism on $n$ tokens can be modeled by a directed graph $H$ on $[n] = \{1, 2, \ldots, n\}$, where a directed edge from $i$ to $j$ indicates a direct interaction between tokens $i$ and $j$, i.e., an inner product that will be computed by the attention mechanism. More precisely, a transformer block can be viewed as a function on the $d$-dimensional embeddings for each of $n$ tokens, mapping from $\mathbb{R}^{d \times n}$ to $\mathbb{R}^{d \times n}$. Let $\mathbf{X} = (\mathbf{x}_1, \mathbf{x}_2, \ldots, \mathbf{x}_n) \in \mathbb{R}^{d \times n}$. A generalized (dot-product) attention mechanism $\text{ATTN}_H : \mathbb{R}^{d \times n} \to \mathbb{R}^{d \times n}$ with attention pattern given by $H$ is defined by

$$\text{ATTN}_H(\mathbf{X})_{:,i} = \mathbf{x}_i + \sum_{j=1}^{h} \mathbf{W}_O^j \mathbf{W}_V^j \mathbf{X}_{\mathcal{N}_H(i)} \cdot \sigma \left( \left( \mathbf{W}_K^j \mathbf{X}_{\mathcal{N}_H(i)} \right)^T \left( \mathbf{W}_Q^j \mathbf{x}_i \right) \right),$$

where $h$ is the number of heads and $m$ is the head size, while $\mathbf{W}_K^j, \mathbf{W}_Q^j, \mathbf{W}_V^j \in \mathbb{R}^{m \times d}$ and $\mathbf{W}_O^j \in \mathbb{R}^{d \times m}$. (The subscript $K$ is for "keys," $Q$ for "queries," $V$ for "values," and $O$ for "output.") Here $\mathbf{X}_{\mathcal{N}_H(i)}$ denotes the submatrix of $\mathbf{X}$ obtained by picking out only those columns corresponding to elements of $\mathcal{N}_H(i)$, the neighbors of $i$ in $H$. We can see that the total number of inner product computations for all $i \in [n]$ is given by the number of edges of $H$. A (generalized) *transformer block* consists of $\text{ATTN}_H$ followed by a feedforward layer:

$$\text{FF}(\mathbf{X}) = \text{ATTN}_H(\mathbf{X}) + \mathbf{W}_2 \cdot \text{ReLU}(\mathbf{W}_1 \cdot \text{ATTN}_H(\mathbf{X}) + \mathbf{b}_1 \mathbf{1}_n^T) + \mathbf{b}_2 \mathbf{1}_n^T,$$

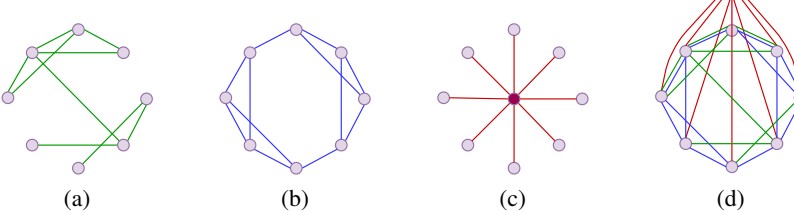

Figure 1: The components of EXPHORMER: (a) shows local neighborhood attention, i.e., edges of the input graph. (b) shows an expander graph with degree 3. (c) shows global attention with a single virtual node. (d) All of the aforementioned components are combined into a single interaction graph that determines the attention pattern of EXPHORMER.

where $\mathbf{W}_1 \in \mathbb{R}^{r \times d}$, $\mathbf{W}_2 \in \mathbb{R}^{d \times r}$, $\mathbf{b}_1 \in \mathbb{R}^r$, and $\mathbf{b}_2 \in \mathbb{R}^d$.

In the standard setting, the $n$ tokens are part of a sequence (e.g., language applications). However, we are concerned with the *graph transformer* setting in which the tokens are nodes of some underlying graph $G = (V, E)$ with $V = [n]$. The attention computation is nearly identical, except that one can also optionally augment it with edge features, as is done in SAN (Kreuzer et al., 2021):

$$\text{ATTN}_H(\mathbf{X})_{:,i} = \mathbf{x}_i + \sum_{j=1}^{h} \mathbf{W}_O^j \mathbf{W}_V^j \mathbf{X}_{\mathcal{N}_H(i)} \cdot \sigma \left( \left( \mathbf{W}_E^j \mathbf{E}_{\mathcal{N}_H(i)} \odot \mathbf{W}_K^j \mathbf{X}_{\mathcal{N}_H(i)} \right)^T \left( \mathbf{W}_Q^j \mathbf{x}_i \right) \right),$$

where $\mathbf{W}_E^j \in \mathbb{R}^{m \times d_E}$, $\mathbf{E}_{\mathcal{N}_H(i)}$ is the $d_E \times |\mathcal{N}_H(i)|$ matrix whose columns are $d_E$-dimensional edge features for the edges connected to node $i$, and $\odot$ denotes element-wise multiplications.

The most typical cases of graph transformers use full (dense) attention, where every token attends to every other node: $H$ is the fully-connected directed graph. As this results in computational complexity $O(n^2)$ for the transformer block, which is prohibitively expensive for large graphs, we wish to replace full attention with a *sparse* attention mechanism, where $H$ has $o(n^2)$ edges – ideally, $O(n)$.

A number of sparse attention mechanisms have been proposed to address the aforementioned issue (see Tay et al., 2020), but the vast majority are designed specifically for functions on *sequences*. EXPHORMER, on the other hand, is a *graph-centric* sparse attention mechanism that makes use of the underlying structure of the input graph $G$. As we will see, EXPHORMER is perhaps most similar in design to the BIGBIRD architecture (Zaheer et al., 2020) designed for functions on sequences, but takes advantage of the underlying graph structure.

We can either use EXPHORMER layers in a pure graph transformer model, or in combination with a message passing network using the GraphGPS framework.

### 3.2 THE EXPHORMER ARCHITECTURE

We now describe the details of the construction of EXPHORMER, an expander-based sparse attention mechanism for graph transformers with $O(|V| + |E|)$ computation, where $G = (V, E)$ is the underlying input graph. The EXPHORMER architecture constructs an interaction graph $H$ that consists of three main components, as shown in Figure 1. The construction always has bidirectional edges, and so $H$ can be viewed as an undirected graph. The mechanism uses three types of edges:

1. **Expander graph attention**: The main building block of our architecture is the use of edges from a random *expander graph*, described in more detail shortly. These graphs have several useful theoretical properties related to spectral approximation and random walk mixing (see Section 4), which allow propagating information between pairs of nodes that are distant in the input graph $G$ without connecting all pairs of nodes. In particular, we use a regular expander graph of constant degree, which allows the number of edges to be just $O(|V|)$. The process we use to construct a random expander graph is described in Appendix C.

2. **Local neighborhood attention**: Another desirable property to capture is *locality*. Graphs carry much more topological structure than sequences, and the neighborhoods of individual

nodes carry a lot of information about connectivity. Thus, we model local interactions by allowing each node $v$ to attend to every other node that is an immediate neighbor of $v$ in $G$: that is, $H$ includes the input graph edges $E$ as well as their reverses, introducing $O(|E|)$ interaction edges. One generalization would be to allow direct attention within $k$-hop neighborhoods, but this might introduce a superlinear number of interactions on general graphs.

3. **Global attention**: The final component is *global attention*, whereby a small number of virtual nodes are added to the interaction graph, and each such node is connected to all the non-virtual nodes. These nodes enable a global "storage sink" and help prove universality properties of EXPHORMER. We will generally add a constant number of virtual nodes, in which case the total number of edges due to global attention will be $O(|V|)$.

The model uses learnable embeddings for expander and global connection edge features, and virtual nodes features. Dataset edge features are used for the local neighborhood edge features.

**Remark 3.1** EXPHORMER *has some conceptual similarities with BigBird, as mentioned previously. For instance, we also make use of virtual global attention nodes, corresponding to* BIGBIRD-ETC.

*However, our approach departs from that of BigBird in some important ways. While BigBird uses $w$-width "window attention" to capture locality of reference, we use local neighborhood attention to capture locality and graph topology. In particular, the interaction graph due to window attention in BigBird can be viewed as a Cayley graph on $\mathbb{Z}_n$, which is sequence-centric, while EXPHORMER is graph-centric and, therefore, uses the structure of the input graph itself to capture locality. BigBird, as implemented for graphs by Rampásek et al. (2022), instead simply orders the graph nodes in an arbitrary sequence and uses windows within that sequence.*

*Both BigBird and* EXPHORMER *also make use of a random attention model. While BigBird uses an Erdős-Rényi graph on $|V|$ nodes, our approach is to use a $d$-regular expander for fixed constant $d$. The astute reader may recall that a Erdős-Rényi graph $G(n,p)$ has spectral expansion properties for large enough $p$. However, it is known that $p = \frac{\log n}{n}$ is the connectivity threshold, i.e., for $p < (1-\epsilon)\frac{\log n}{n}$, $G(n,p)$ is almost surely a disconnected graph. Therefore, in order to obtain even a connected graph in the Erdős-Rényi model – let alone one with expansion properties – one would need $p = \Omega\left(\frac{\log n}{n}\right)$, giving superlinear complexity for the number of edges. BigBird uses $p = \Theta(1/n)$, keeping a linear number of edges but losing expansion properties. Our expander graphs, by contrast, allow both a linear number of edges and guaranteed spectral expansion properties.*

*We will see in the practical experiments of Section 5 that* EXPHORMER*-based models often substantially outperform BigBird-based equivalents, with fewer parameters.*

**Remark 3.2** *Previous graph-oriented transformers have, naturally, used the graph structure in their attention mechanisms. The SAN architecture (Kreuzer et al., 2021) uses two attention mechanisms, one based on the input edges $E$ and one based on all the other edges not present in the input graph. By using a single attention mechanism,* EXPHORMER*s introduce fewer additional parameters, allowing for more compact models that also significantly outperform SAN in our experiments.*

## 4 THEORETICAL PROPERTIES OF EXPHORMER

EXPHORMER is based on expander graphs, which have a number of properties that make them suitable as a key building block of our approach. In this section, we describe relevant properties of expander graphs along with their implications for EXPHORMERs.

### 4.1 BASICS OF EXPANDER GRAPHS AND LAPLACIANS

For simplicity, let us consider $d-$*regular* graphs (where every node has $d$ neighbors). Suppose $G$ is a $d$-regular undirected graph on $n$ vertices. Let $A_G$ be the $n \times n$ adjacency matrix of $G$. It is known that $A_G$ has $n$ real eigenvalues $d = \lambda_1 \geq \lambda_2 \geq \cdots \geq \lambda_n \geq -d$. The graph $G$ is said to be an $\epsilon$-*expander* if $\max\{|\lambda_2|, |\lambda_n|\} \leq (1-\epsilon)d$ (Alon, 1986).

Intuitively speaking, expander graphs are sparse approximations of complete graphs. It is known that expanders have several useful properties (viz., edge expansion and mixing properties), which make the graph "well-connected". A higher $\epsilon$ corresponds to better expansion properties.

For example, in an expander graph, the diameter, $O(\log_d n)$, is as low as possible for a given number of edges without a bottleneck cut (Alon, 1986), which is defined as:

$$\left| E \cap S \times \overline{S} \right| \geq (1 - \epsilon) \frac{d|S||\overline{S}|}{n}. \tag{1}$$

## 4.2 Expander Graphs as Approximators of Complete Graphs

We now outline some important properties of an expander-based attention mechanism. Roughly speaking, our goal is to replace a densely connected graph, i.e., a complete graph, with $\Theta(n^2)$ edges by a graph with $o(n^2)$, preferably $O(n)$, edges that preserves certain properties of the complete graph. We show that suitable expander graphs, in fact, achieve this.

### 4.2.1 Spectral Properties

A useful tool to study expanders is the *Laplacian* matrix of a graph. Letting $D_G$ denote the $n \times n$ diagonal matrix whose $i$-th diagonal entry is the degree of the $i$-th node, we define $L_G = D_G - A_G$ to be the Laplacian of $G$. It is known that $L_G$ captures several important spectral properties of the graph. The first useful property of complete graphs that expander graphs (approximately) preserve is the spectral decomposition of the Laplacian — per this well known theorem in spectral graph theory.

**Theorem 4.1** *A $d$-regular $\epsilon$-expander $G$ on $n$ vertices spectrally approximates the complete graph $K_n$ on $n$ vertices, i.e.,* [1]

$$(1 - \epsilon)\frac{1}{n}L_K \preceq \frac{1}{d}L_G \preceq (1 + \epsilon)\frac{1}{n}L_K.$$

Spectral approximation is known to preserve the cut structure in graphs. As a result, replacing a dense attention mechanism with a sparse attention mechanism along the edges of an expander graph retains spectral properties (viz., cuts, vertex expansion, etc.) .

### 4.2.2 Mixing Properties

Another property of expanders is that random walks mix well. Let $G = (V, E)$ be a $d$-regular $\epsilon$-expander. Consider a random walk $v_0, v_1, v_2, \ldots$ on $G$, where $v_0$ is chosen according to an initial distribution $\pi^{(0)}$, and then each subsequent $v_{t+1}$ is one of the $d$ neighbors of $v_t$ chosen uniformly at random. Then each $v_t$ is distributed according to the probability distribution $\pi^{(t)} : V \to \mathbb{R}^+$, given recursively by $\pi^{(t+1)} = D_G^{-1} A_G \pi^{(t)}$. It turns out that after a logarithmic number of steps, a random walk from a starting probability distribution on the vertices is close to uniformly distributed along all nodes of the graph.

**Lemma 4.2** *(Alon, 1986) Let $G = (V, E)$ be a $d$-regular $\epsilon$-expander graph on $n = |V|$ nodes. For any initial distribution $\pi^{(0)} : V \to \mathbb{R}^+$ and any $\delta > 0$, $\pi^{(t)} : V \to \mathbb{R}^+$ satisfies*

$$\|\pi^{(t)} - 1/n\|_1 \leq \delta,$$

*as long as $t = \Omega\left(\frac{1}{\epsilon}\log(n/\delta)\right)$, i.e., the resulting distribution over the nodes of $G$ of a random walk with $t$ walks starting from the distribution $\pi^{(0)}$ is $\delta$-close in $L^1$-norm to the uniform distribution.*

In an attention mechanism of a transformer, one can consider the graph of pairwise interactions (i.e., $i$ is connected to $j$ if $i$ and $j$ attend to each other). If the attention mechanism is dense, then each node is connected to every other node and it is trivial to see that every pair of nodes interacts with each other in a single transformer layer. In a *sparse* attention mechanism, on the other hand, some pairs of nodes are not directly connected, meaning that a single transformer layer will not model interactions between all pairs of nodes. However, if we stack transformer layers on top of each

---

[1]Notation: given matrices $A$ and $B$, we say that $A \preceq B$ if $B - A$ is a positive semi-definite matrix.

other, the stack will be able to model longer range interactions. In particular, a consequence of the above lemma is that if our sparse attention mechanism is modeled after an $\epsilon$-expander graph, then stacking at least $t = \frac{1}{\epsilon}\log(n/\delta)$ layers will model "most" pairwise interactions between nodes.

A related property concerns the *diameter* of expander graphs. One can, in fact, show that the diameter of a regular expander graph is logarithmic in the number of nodes, asymptotically.

**Theorem 4.3** *(Alon, 1986) Suppose $G = (V, E)$ is a $d$-regular $\epsilon$-expander graph on $n$ vertices. Then, for every vertex $v$ and $k \geq 0$, the $k$-hop neighborhood $B(v, r) = \{w \in V : d(v, w) \leq k\}$ has*

$$|B(v, r)| \geq \min\{(1+c)^k, n\}$$

*for some constant $c > 0$ depending on $d, \epsilon$. In particular, we have that $\mathrm{diam}(G) = O_{d,\epsilon}(\log n)$.*

As a consequence, we obtain the following result, which shows that using logarithmically many successive transformer layers allows each node to propagate information to every node.

**Corollary 4.4** *If a sparse attention mechanism on $n$ nodes is modeled as a $d$-regular $\epsilon$-expander graph, then stacking $O_{d,\epsilon}(\log n)$ transformer layers models all pairwise node interactions.*

### 4.3 EXPHORMER-BASED TRANSFORMERS ARE UNIVERSAL APPROXIMATORS

While the expander graph component of EXPHORMER guarantees that $O(\log n)$ graph transformer layers are enough to allow each node to interact with every other node, it may still be desirable to enable node interactions with a smaller number of layers (e.g., $O(\log n)$ can still be infeasible when the number of nodes, $n$, is extremely large). The global attention component allows this by essentially serving as a "short circuit" by which every node can interact with every other node using just two graph transformer layers.

The global attention component also helps us to obtain a universal approximation property of EX-PHORMER. In particular, continuous functions $f : [0, 1]^{d \times |V|} \to \mathbb{R}^{d \times |V|}$ can be approximated to desired accuracy by an EXPHORMER network as long as there is at least one virtual node. We defer the details to the appendix (see Appendix D).

## 5 EXPERIMENTS

In this section, we evaluate the empirical performance of graph transformer models based on EXPHORMER on a wide variety of graph datasets with graph prediction and node prediction tasks Dwivedi et al. (2020); Hu et al. (2020); Freitas et al. (2021). We perform ablation studies on eight benchmark datasets, including image-based graph datasets (CIFAR10, MNIST), a molecular dataset (ogbg-molpcba), synthetic datasets (PATTERN, CLUSTER), and datasets based on code graphs (ogbg-code2, MalnetTiny). We also demonstrate the use of EXPHORMER on a large citation

| Model | CIFAR10 Accuracy ↑ | MalNet-Tiny Accuracy ↑ | MNIST Accuracy ↑ | CLUSTER Accuracy ↑ | PATTERN Accuracy ↑ |
|---|---|---|---|---|---|
| GCN Kipf & Welling (2017) | 55.710±0.381 | 81.0 | 90.705±0.218 | 68.498 ± 0.976 | 71.892 ± 0.334 |
| GIN Xu et al. (2018) | 55.255±1.527 | 88.98±0.557 | 96.485±0.252 | 64.716 ± 1.553 | 85.387 ± 0.136 |
| GAT Veličković et al. (2018) | 64.223±0.455 | 92.1 ±0.242 | 95.535±0.205 | 70.587 ± 0.447 | 78.271 ± 0.186 |
| GatedGCN Bresson & Laurent (2017); Dwivedi et al. (2020) | 67.312±0.311 | 92.23±0.65 | 97.340±0.143 | 73.840 ± 0.326 | 85.568 ± 0.088 |
| PNA Corso et al. (2020) | 70.35±0.63 | – | 97.94±0.12 | – | – |
| DGN Beaini et al. (2021) | 72.838±0.417 | – | – | – | 86.680±0.034 |
| CRaWl Toenshoff et al. (2021) | 69.013±0.259 | – | 97.944±0.050 | – | – |
| GIN-AK+ Zhao et al. (2022) | 72.19±0.13 | – | – | – | 86.850±0.057 |
| SAN Kreuzer et al. (2021) | – | – | – | 76.691±0.65 | 86.581±0.037 |
| K-Subgraph SAT Chen et al. (2022) | – | – | – | 77.856±0.104 | 86.848±0.037 |
| EGT-SPE(+DO) Hussain et al. (2021) | 67.004±0.624 | – | 97.722±0.222 | 77.909±0.245 | 86.730±0.036 |
| GraphGPS Rampášek et al. (2022) | 72.298±0.356 | 93.50±0.41 | 98.051±0.126 | 77.95±0.305 | 90.324±0.132 |
| EXPHORMER | 72.884±0.166 | 92.24±0.291 | 98.238 ± 0.039 | 77.295 ± 0.060 | 88.079±0.104 |
| MPNN+EXPHORMER | 74.754±0.194 | 93.16±0.137 | 98.414 ± 0.038 | 78.024±0.041 | 90.522±0.030 |

Table 1: Comparison of EXPHORMER with baselines on various datasets. Best results are colored in **first**, **second**, **third**.

| Model | #Layers | Hidden layers | #Positional encoding | Expander degree | #Parameters | Time Epch/Total | Accuracy |
|---|---|---|---|---|---|---|---|
| GPS-MPNN: GatedGCN | 3 | 52 | LapPE | - | 79,654 | 43s/1.18h | $69.95 \pm 0.499$ |
| GPS: Transformer | 3 | 52 | LapPE | - | 70,762 | 40s/1.11h | $68.86 \pm 1.138$ |
| GPS: Transformer + MPNN | 5 | 40 | ESLapPE | - | 111,735 | 104s/2.89h | $73.53 \pm 0.238$ |
| GPS: Transformer + MPNN | 3 | 52 | LapPE | - | 112,726 | 62s/1.72h | $72.31 \pm 0.344$ |
| GPS: Performer + MPNN | 5 | 40 | ESLapPE | - | 283,935 | 132s/3.65h | $70.18 \pm 0.095$ |
| GPS: Performer + MPNN | 3 | 52 | LapPE | - | 239,554 | 77s/2.14h | $70.67 \pm 0.338$ |
| GPS: BigBird + MPNN | 5 | 40 | ESLapPE | - | 128,335 | 243s/6.75h | $70.51 \pm 0.256$ |
| GPS: BigBird + MPNN | 3 | 52 | LapPE | - | 129,418 | 145s/4h | $70.48 \pm 0.106$ |
| EXPHORMER | 5 | 44 | ESLapPE | 5 | 84,134 | 64s/1.78h | $72.33 \pm 0.155$ |
| EXPHORMER | 7 | 44 | ESLapPE | 5 | 119,022 | 80s/2.23h | $72.88 \pm 0.166$ |
| EXPHORMER + MPNN | 5 | 40 | ESLapPE | 5 | 111,095 | 115s/3.21h | $74.75 \pm 0.194$ |
| EXPHORMER + MPNN | 5 | 44 | ESLapPE | 5 | 133,819 | 114s/3.19h | $\mathbf{75.03 \pm 0.186}$ |

Table 2: Results with varying attention and MPNNs on CIFAR10. EXPHORMER with MPNN provides the highest accuracy. Also, pure transformer models based on EXPHORMER (without the use of MPNNs) are comparable.

network, ogbn-arxiv. A full description of the datasets is found in A. Experiments for ogbn-arxiv have been performed on NVIDIA A100 GPUs, while all other experiments were on NVIDIA V100s.

## 5.1 COMPARISON OF EXPHORMER-BASED MODELS TO BASELINES

We apply EXPHORMER to the modular GraphGPS framework (Rampásek et al., 2022), which constructs graph transformer models by composing attention mechanisms with message-passing schemes, together with a choice of positional and structural encodings. We also show the results for *pure attention* EXPHORMER-only models that do not use a message passing network. Table 1 shows results on five datasets from the Benchmarking GNNs collection (Dwivedi et al., 2020), along with MalNet-Tiny (Freitas et al., 2021). Using an EXPHORMER-based graph transformer with message-passing in the GraphGPS framework yields competitive results, including *state-of-the-art (SOTA) on three of the datasets*. Notably, our EXPHORMER models outperform not only the MPNN baselines but also recent *full (dense) attention* graph transformer models (i.e., SAN and full transformer models in the GraphGPS paper). Tables 2 and 3 show the EXPHORMER architecture have are significantly fewer training parameters but provide much better accuracy, suggesting that the structured sparsity in EXPHORMER can offer regularization advantages as well as time and memory scalability.

Table 5 (see Appendix B.1) gives similar comparisons for two datasets from the OGB collection Hu et al. (2020): ogbg-molpcba and ogbg-code2. Our models are competitive with existing baselines. Our mildly worse performance on ogb-molpcba is perhaps not surprising, given that this is a molecular dataset consisting of small graphs (an average of 26 nodes). On small graphs, the attention pattern in EXPHORMER is unlikely to be very different from that of the dense attention mechanism; thus the scope for us to improve is limited. Indeed, the greatest improvements from EXPHORMER are on datasets where the average number of nodes is larger, e.g., around 100 or more nodes.

## 5.2 COMPARISON OF ATTENTION MECHANISMS

We now discuss a series of experiments to isolate the performance of pure attention EXPHORMER model compared to other attention mechanisms, which will help isolate the attention mechanism used in the model. For each dataset, we take our EXPHORMER-based models and compare them to models obtained by replacing the EXPHORMER attention mechanism with other dense (full transformer) and sparse (BigBird, Performer) attention mechanisms. As different mechanisms can require drastically different numbers of parameters, we train two models per mechanism: one in which all hyperparameters are kept the same but vary the attention mechanisms; and another in which the hyperparameters are adjusted in order to keep the total number of parameters of the model roughly the same.

Results for the CIFAR10 dataset are in Table 2, the other datasets are in the Appendix. EXPHORMER-based GPS models (with both EXPHORMER attention and MPNNs) outperform comparable models where the attention mechanism is replaced (by full attention or a sparse attention model such as BigBird and Performer). Similar comparisons on other datasets are in Appendix B.3; again, EXPHORMER outperforms other sparse attention mechanisms while either surpassing or performing comparably to full (dense) attention.

## 5.3 PURE TRANSFORMER ARCHITECTURES

The results presented thus far naturally lead to questions: (1.) how much of the performance gain of EXPHORMER-based GPS models is attributable to the attention component as opposed to the MPNN component, and (2.) whether "pure" *sparse* transformer models can achieve good performance by itself. Indeed, Rampásek et al. (2022) present ablation studies showing that removing the MPNN component from the proposed GraphGPS models results in significantly worse performance. Similarly, Kreuzer et al. (2021) offer two variants of their SAN architecture, the "full" variant that uses dense attention, as well as a "sparse" variant that allows nodes to attend only to direct neighbors in the input graph—their ablation results show that their "sparse" variant performs significantly worse than the "full" variant, often underperforming pure MPNNs.

In order to address these questions, we also train *pure* EXPHORMER-based sparse transformer models that use *only attention without any message passing*. The results for CIFAR10 in Table 2 show that the pure EXPHORMER models (labeled "Exphormer"), in fact, outperform pure transformer GPS models as well as most GPS models that use MPNNs. The main exception is the dense attention "Transformer+MPNN" model; however our pure EXPHORMER model performs competitively. Similar comparisons on other datasets are shown in Appendix B.3.

## 5.4 SCALABILITY TO LARGE-SCALE GRAPHS

One of the difficulties with graph transformer architectures has been their poor scalability to larger graphs with thousands of nodes. Dense attention mechanisms, like SAN and Graphormer, with quadratic memory complexity and time complexity restrict their applicability to datasets on graphs with a small number of nodes.

GraphGPS Rampásek et al. (2022) used sparse attention mechanisms but their architecture handles graphs of up to about 5,000 nodes, in MalNet-Tiny (Freitas et al., 2021). Again, EXPHORMER-based models provide improved accuracy, as shown in Tables 1 and 10.

Our work allows us to extend graph transformer architectures to far larger graphs, with hundreds of thousands of nodes. We show that EXPHORMER architecture can scale, with competetive accuracy, to ogbn-arxiv (Hu et al., 2020), a transductive dataset consisting of a single large graph of the citation network of over 160K arXiv papers, containing over a million edges. Specifically, we achieve a test accuracy of **0.7196** using the EXPHORMER architectures. At the time of writing, a relevant leaderboard ows 0.7637 as the highest reported test accuracy, based on adaptive graph diffusion networks (Zhao et al., 2021). Table 3 shows the relative performance of EXPHORMER compared to other Transformers. A dense full transformer does not even fit in memory on an NVidia A100 GPU (even with only 2 layers and 32 hidden dimensions). The best accuracy for other sparse models was found with networks of 3 hidden layers and 96 hidden dimensions. Notice that BigBird and Performer have significantly longer epoch times and worse performance compared to EXPHORMER with degree 3 expander edges.

| Model | Accuracy | Epoch times | #Parameters |
|---|---|---|---|
| BigBird | $55.19 \pm 0.16$ | 16.745 | 296,968 |
| Performer | $55.24 \pm 0.05$ | 8.1905 | 276,808 |
| EXPHORMER | $71.51 \pm 0.04$ | 2.097 | 267,976 |
| EXPHORMER+GCN | $71.96 \pm 0.12$ | 0.9652 | 268,264 |

Table 3: Comparison of attention mechanisms on the ogbn-arxiv dataset.

## 6 CONCLUSION

EXPHORMER is a new sparse graph transformer architecture built on expander graphs. We have shown that the relevant mathematical properties of expander graphs make EXPHORMER a suitable choice for graph learning, with time and memory complexity linear in the size of the graph. Using EXPHORMER in the GraphGPS framework allows us to obtain state-of-the-art empirical results on a number of datasets while also allowing graph transformers to scale to datasets on large graphs, a realm which has proved elusive for graph transformers in the past.

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

# A  DATASET DESCRIPTIONS

Below, we provide descriptions of the datasets on which we conduct experiments.

**CIFAR10 and MNIST**   CIFAR10 and MNIST are the graph equivalents of the eponymous image classification datasets. A graph is created by constructing the 8-nearest neighbor graph of the SLIC superpixels of the image. These are both. 10-class classification problems.

**CLUSTER and PATTERN**   PATTERN and CLUSTER are node classification problems. Both are synthetic datasets that are sampled from the Stochastic Block Model (SBM), which is a popular way to model communities. In PATTERN, the prediction task is to identify if a node belongs to one of the 100 possible predetermined subgraph patterns. In CLUSTER, the goal is to identify the cluster to which a node belongs to, when the nodes are selected from 6 different clusters with the same distribution.

**MalNet-Tiny**   Malnet-Tiny is a smaller dataset generated from a larger dataset for identifying malware based on function call graphs. The dataset contains the call graphs from the Android APKs. The tiny dataset contains 5000 graphs. The graphs can have up to 5000 nodes. The task is to predict the graph as being benign or from one of four types of malware.

**ogbg-code2**   The ogbg-code2 dataset has the abstract sytax trees (ASTs) of python methods curated from github. The prediction problem is "code sumarization". To determine the name of a function given the body of the method.

**ogbg-molpcba**   ogbg-molpcba is a molecular datasets. It has graphs representing molecules, where the nodes and edges represent the atoms and their chemical bonds, respectively. The node features include atomic number, chirality etc. The goal is to predict binary labels. The dataset is heavily skewed as only only 1.4% of data has positive labels.

**ogbn-arxiv**   The ogbn-arxiv dataset consists of one large directed graph of 169343 nodes and 1,166,243 edges representing a citation network between all computer science papers on arXiv that were indexed by the Microsoft academic graph. Nodes in the graph represent papers, while a directed edge indicates that a paper cites another. Each node has an 128-dimensional feature vector derived from embeddings of words in the title and abstract of the underlying paper. The prediction task is a 40-class node classification problem — to identify the primary category of each arXiv paper, as listed by the authors. Moreover, the nodes of the citation graph are split into 90K training nodes, 30K validation notes, and 48K test nodes.

Table 4 shows a summary of the statistics of the aforementioned datasets.

Table 4: Dataset statistics

| Dataset | Graphs | Avg. nodes | Avg. edges | Prediction Level | No. Classes | Metric |
|---|---|---|---|---|---|---|
| MNIST | 70,000 | 70.6 | 564.5 | graph | 10 | Accuracy |
| CIFAR10 | 60,000 | 117.6 | 941.1 | graph | 10 | Accuracy |
| PATTERN | 14,000 | 118.9 | 3,039.3 | inductive node | 2 | Accuracy |
| CLUSTER | 12,000 | 117.2 | 2,150.9 | inductive node | 6 | Accuracy |
| ogbg-molpcba | 437,929 | 26.0 | 28.1 | graph | 128 | Avg. Precision |
| ogbg-code2 | 452,741 | 125.2 | 124.2 | graph | 5 token sequence | F1 score |
| MalNet-Tiny | 5,000 | 1,410.3 | 2,859.9 | graph | 5 | Accuracy |
| ogbn-arxiv | 1 | 169,343 | 1,166,243 | node | 40 | Accuracy |

# B  MORE EXPERIMENTAL RESULTS

## B.1  BENCHMARKS ON OGB DATASETS

In Table 5 we show the results for OGB datasets (see the discussion in Section 5.1).

| Model | ogbg-molpcba Avg. Precision ↑ | ogbg-code2 F1 score ↑ |
|---|---|---|
| GCN+virtual node | 0.2424 ± 0.0034 | 0.1595 ± 0.0018 |
| GIN+virtual node | 0.2703 ± 0.0023 | 0.1581 ± 0.0026 |
| GatedGCN-LSPE | 0.267 ± 0.002 | – |
| PNA | 0.2838 ± 0.0035 | 0.1570 ± 0.0032 |
| DeeperGCN | 0.2781 ± 0.0038 | – |
| DAGNN | – | 0.1751 ± 0.0049 |
| DGN | 0.2885 ± 0.0030 | – |
| GIN-AK+ | 0.2930 ± 0.0044 | – |
| CRaWl | 0.2986 ± 0.0025 | – |
| ExpC Yang et al. (2022) | 0.2342 ± 0.0029 | – |
| SAN | 0.2765 ± 0.0042 | – |
| GraphTrans (GCN-Virtual) | 0.2761 ± 0.0029 | 0.1830 ± 0.0024 |
| K-Subtree SAT | – | 0.1937 ± 0.0028 |
| GraphGPS | 0.2907 ± 0.0028 | 0.1894 ± 0.0024 |
| MPNN+Exphormer | 0.2823 ± 0.0004 | 0.18507 ± 0.047 |

Table 5: Comparison of EXPHORMER with baselines on various datasets from the Open Graph Benchmark (OGB) (Hu et al., 2020)

## B.2 HYPERPARAMETERS

Our hyperparameter choices, including the optimizer, positional encodings, and structural encodings, were guided by the instructions in GraphGPS (Rampásek et al., 2022). There were some cases, however, when more layers with smaller dimensions gave better results in EXPHORMER. This may be due to the fact that each node gets fewer inputs for each layer, but EXPHORME requires more layers in order to propagate well. Additionally, we observed that Equivstable Laplacian Positional Encoding (ESLapPE) performed better than normal Laplacian Positional Encoding (LapPE). In our experiments, it consistently replaced LapPE. For the GPS models with Exphormer, we consistently use GatedGCN except for the ogbn-arxiv dataset, which we use GCN.

Through our model, some extra hyperparameters are introduced — the degree of the graph expander and the number of virtual nodes. For these hyperparameters, we used linear search and found that expander degree 3-5 was the most effective. Depending on the graph size, we used 1-5 virtual nodes.

To make fair comparisons, we used a similar parameter-budget to GraphGPS. For PATTERN and CLUSTER, we used a parameter-budget of 500K, and for CIFAR and MNIST, we used a parameter-budget of around 100K. See details in Table 6 and Table 8.

| Hyperparameter | CIFAR10 | MNIST | MalNet-Tiny | PATTERN | CLUSTER |
|---|---|---|---|---|---|
| Num Layers | 5 | 5 | 5 | 3 | 16 |
| Hidden Dim | 40 | 40 | 64 | 84 | 48 |
| Num Heads | 4 | 4 | 4 | 4 | 8 |
| Dropout | 0.1 | 0.1 | 0.15 | 0.0 | 0.0 |
| PE | ESLapPE | ESLapPE | None | ESLapPE | ESLapPE |
| Batch Size | 16 | 16 | 16 | 32 | 16 |
| Learning Rate | 0.001 | 0.001 | 0.0005 | 0.001 | 0.00001 |
| Num Epochs | 150 | 150 | 150 | 100 | 100 |
| Expander Degree | 5 | 5 | 5 | 5 | 5 |
| Num Virtual Nodes | 1 | 1 | 1 | 4 | 5 |
| Num parameters | 111,095 | 111,015 | 285,071 | 297,280 | 502,102 |

Table 6: Hyperparameters used for GPS model using EXPHORMER for datasets: CIFAR10, MNIST, MalNet-Tiny, PATTERN, CLUSTER.

## B.3 FULL COMPARISON OF ATTENTION MECHANISMS

In Section 5.2, we presented two approaches for the comparison of models trained using different attention mechanisms — fixing the hyperparameters and fixing a budget on the total number of trainable parameters. The results showed the advantage of EXPHORMER over other attention mech-

| Hyperparameter | ogbg-molpcba | ogbg-code2 |
|---|---|---|
| Num Layers | 5 | 4 |
| Hidden Dim | 40 | 256 |
| Num Heads | 4 | 4 |
| Dropout | 0.2 | 0.2 |
| PE | RWSE | ASTNode |
| Batch Size | 512 | 32 |
| Learning Rate | 0.0005 | 0.0001 |
| Num Epochs | 100 | 30 |
| Expander Degree | 5 | 5 |
| Num Virtual Nodes | 5 | 0 |
| Num parameters | 9,742,960 | 12,453,298 |

Table 7: Hyperparameters used for GPS model using EXPHORMER for datasets: ogbg-molpcba, ogbg-code2.

| Hyperparameter | CIFAR10 | MNIST | MalNet-Tiny | PATTERN | CLUSTER |
|---|---|---|---|---|---|
| Num Layers | 7 | 7 | 5 | 4 | 16 |
| Hidden Dim | 44 | 44 | 64 | 84 | 48 |
| Num Heads | 4 | 4 | 4 | 4 | 8 |
| Dropout | 0.1 | 0.1 | 0.15 | 0.0 | 0.1 |
| PE | ESLapPE | ESLapPE | None | ESLapPE | ESLapPE |
| Batch Size | 16 | 16 | 16 | 32 | 16 |
| Learning Rate | 0.001 | 0.001 | 0.0005 | 0.001 | 0.001 |
| Num Epochs | 150 | 150 | 150 | 100 | 150 |
| Expander Degree | 5 | 5 | 5 | 5 | 5 |
| Num Virtual Nodes | 1 | 1 | 1 | 5 | 2 |
| Num parameters | 119,022 | 127,698 | 285,071 | 273,757 | 344,214 |

Table 8: Hyperparameters used for EXPHORMER model (without MPNN) for datasets: CIFAR10, MNIST, MalNet-Tiny, PATTERN, CLUSTER.

anisms for CIFAR10 (Table 2) and PATTERN (Table 11). Here, we present similar results for the remaining datasets — MNIST in Table 9; MALNET-Tiny in Table 10; PATTERN in Table 11; and CLUSTER in Table 12.

| Model | #Layers | Hidden layers | #Positional encoding | Expander degree | #Parameters | Time Epoch/Total | Accuracy |
|---|---|---|---|---|---|---|---|
| GPS: Transformer + MPNN | 5 | 40 | ESLapPE | - | 111,655 | 131s/5.45h | $98.336 \pm 0.0189$ |
| GPS: Transformer + MPNN | 3 | 52 | LapPE | - | 115,394 | 76s/2.13h | $98.051 \pm 0.126$ |
| GPS: Performer + MPNN | 5 | 40 | ESLapPE | - | 283,855 | 156s/6.52h | $98.34 \pm 0.0349$ |
| GPS: BigBird + MPNN | 5 | 40 | ESLapPE | - | 128,255 | 267s/11.11h | $98.176 \pm 0.0146$ |
| EXPHORMER | 5 | 44 | ESLapPE | 5 | 92,146 | 75s/3.14h | $98.08 \pm 0.051$ |
| EXPHORMER | 7 | 44 | ESLapPE | 5 | 127,698 | 93s/3.87h | $98.238 \pm 0.0387$ |
| EXPHORMER + MPNN | 5 | 40 | ESLapPE | 5 | 111,015 | 132s/5.49h | $98.414 \pm 0.035$ |
| EXPHORMER + MPNN | 5 | 44 | ESLapPE | 5 | 133,731 | 137s/5.72h | $98.424 \pm 0.018$ |

Table 9: Ablation studies results for MNIST

## B.4 EFFECT OF DIFFERENT COMPONENTS OF THE MODEL

[In the final version, this section will include similar analyses of more datasets.]

Here we analyze the effect of each of the components of the model. Our Exphormer model has three main components: local neighborhood, expander edges, and virtual nodes. In Table 13, we can see that removing each component leads to poorer results. The effect of local neighborhood edges is much more important in Exphormer models that do not include an MPNN, suggesting that local

| Model | #Layers | Hidden layers | #Positional encoding | Expander degree | #Parameters | Time Epoch/Total | Accuracy |
|---|---|---|---|---|---|---|---|
| GPS-MPNN: GatedGCN | 5 | 64 | - | - | 199,237 | 6s/0.25h | $92.23 \pm 0.65$ |
| GPS: Performer | 5 | 64 | - | - | 421,957 | 41s/1.73h | $73.90 \pm 0.58$ |
| GPS: Transformer + MPNN* | 5 | 64 | - | - | 282,437 | 94s/3.94h | $93.50 \pm 0.41$ |
| GPS: Performer + MPNN | 5 | 64 | - | - | 527,237 | 46s/1.90h | $92.64 \pm 0.78$ |
| GPS: BigBird + MPNN | 5 | 64 | - | - | 324,357 | 130s/5.43h | $92.34 \pm 0.34$ |
| EXPHORMER | 5 | 80 | - | 5 | 283,173 | 25.2s/1.05h | $92.18 \pm 0.292$ |
| EXPHORMER | 8 | 64 | - | 5 | 296,325 | 35.2s/1.47h | $92.24 \pm 0.291$ |
| EXPHORMER + MPNN | 5 | 64 | - | 5 | 285,701 | 27.5s/1.15h | $93.16 \pm 0.137$ |

Table 10: Ablation studies results for MalNet-Tiny. The model marked with * did not fit in memory with batch size 16, and was trained with batch size 8.

| Model | #Layers | Hidden dimension | #Positional encoding | #Parameters | Time Epoch/Total | Accuracy |
|---|---|---|---|---|---|---|
| GPS: Transformer + MPNN | 3 | 84 | ESLapPE | 297,196 | | $90.313 \pm 0.109$ |
| GPS: Transformer + MPNN | 6 | 64 | LapPE-10 | 337,201 | | $90.324 \pm 0.132$ |
| GPS: Performer + MPNN | 3 | 84 | ESLapPE | 469,816 | | $88.571 \pm 0.144$ |
| GPS: Performer + MPNN | 3 | 56 | ESLapPE | 267,740 | | $88.154 \pm 0.189$ |
| GPS: BigBird + MPNN | 3 | 84 | ESLapPE | 340,288 | | $90.260 \pm 0.167$ |
| GPS: BigBird + MPNN | 3 | 76 | ESLapPE | 279,304 | | $90.446 \pm 0.047$ |
| EXPHORMER + MPNN | 3 | 84 | ESLapPE | 297,280 | | $90.522 \pm 0.030$ |

Table 11: Ablation studies results for PATTERN

and structural encodings are not fully sufficient to capture the structure of the graph and cannot be replace the actual edges.

Table 14 shows the effect of the degree of the expander graph. Selecting the right expander degree is important; results can vary a lot for different expander degrees.

Table 15 shows a similar study on number of virtual nodes.

## C    DETAILS OF EXPANDER GRAPH CONSTRUCTION

A major component of EXPHORMER is the use of the edges of an expander graph. Thus far, we have not specified which expander graph to use. In this section, we provide details of the specific expander graphs we use as well and quantify their spectral expansion properties.

### C.1    RAMANUJAN GRAPHS

A natural question is how strong the spectral expansion properties of a $d$-regular graph can be, i.e., for how large an $\epsilon > 0$ does a $d$-regular $\epsilon$-expander exist. The following theorem gives a bound on how large the spectral gap can be.

**Theorem C.1 (Alon-Boppana)** *Let $d > 0$. The eigenvalues of the adjacency matrix of a $d$-regular graph on $n$ nodes satisfy*

$$\max\{|\lambda_2|, |\lambda_n|\} \geq 2\sqrt{d-1} - o_n(1).$$

*In other words, a $d$-regular $\epsilon$-expander graph can exist only for $\epsilon \leq 1 - \frac{2\sqrt{d-1}}{d} + o_n(1)$.*

| Model | #Layers | Hidden layers | #Positional encoding | #Parameters | Time Epoch/Total | Accuracy |
|---|---|---|---|---|---|---|
| GPS: Transformer + MPNN | 16 | 48 | LapPE-10 | 502,054 | | $77.95 \pm 0.305$ |
| GPS: Performer + MPNN | 16 | 48 | ESLapPE | 1,927,510 | | $78.539 \pm 0.069$ |
| GPS: Performer + MPNN | 4 | 48 | ESLapPE | 486,346 | | $75.91 \pm 0.01$ |
| GPS: BigBird + MPNN | 16 | 48 | ESLapPE | 580,438 | | $77.247 \pm 0.052$ |
| GPS: BigBird + MPNN | 16 | 40 | ESLapPE | 406,262 | | $77.212 \pm 0.089$ |
| EXPHORMER + MPNN | 16 | 48 | ESLapPE | 502,102 | | $78.024 \pm 0.041$ |

Table 12: Ablation studies results for CLUSTER

| Dataset | Model | No Local Neighborhood | No Expander Edges | No Global Connections | All Components |
|---------|-------|----------------------|-------------------|----------------------|----------------|
| Cifar10 | Exphormer | $64.91 \pm 0.199$ | $71.36 \pm 0.205$ | $71.54 \pm 0.112$ | $\mathbf{72.33 \pm 0.155}$ |
| | Exphormer+MPNN | $74.15 \pm 0.143$ | $74.53 \pm 0.189$ | $74.57 \pm 0.183$ | $\mathbf{74.75 \pm 0.194}$ |

Table 13: Results for the full model versus removing each of the components. Each component helps the final model.

| Dataset | Model | Expander Graph Degree | | | | |
|---------|-------|------|------|------|------|------|
| | | 3 | 4 | 5 | 7 | 10 |
| Cifar10 | Exphormer | $71.36 \pm 0.215$ | $71.32 \pm 0.243$ | $\mathbf{72.33 \pm 0.155}$ | $71.91 \pm 0.272$ | $71.96 \pm 0.264$ |
| | Exphormer+MPNN | $74.57 \pm 0.143$ | $74.10 \pm 0.122$ | $\mathbf{74.75 \pm 0.194}$ | $74.54 \pm 0.296$ | $74.70 \pm 0.226$ |

Table 14: Effect of the selection of the expander degree on the results. We can see that correct selecton of the expander degree does affect the quality of the final model.

As it turns out, there exist $\epsilon$-expander graphs with $\epsilon$ achieving this bound. In fact, a $d$-regular $\epsilon$-expander graph satisfying $\epsilon \geq 1 - \frac{2\sqrt{d-1}}{d}$ is known as a *Ramanujan graph*. Ramanujan graphs are essentially the best possible spectral expanders, and several constructions have been considered over the years (Lubotzky et al., 1988; Margulis, 1988).

## C.2   RANDOM REGULAR GRAPHS

While there exist deterministic constructions of Ramanujan graphs, they are often algebraic/number theoretic in nature and therefore exist only for specific choices of $d$ (e.g., the constructions of Lubotzky et al. (1988) as well as independently of Margulis (1988), for which one requires $d \equiv 2 \pmod 4$ and $d - 1$ to be a prime). Recently, the work of Alon (2021) showed a construction of strongly explicit near-Ramanujan graphs of every degree, but it should be noted that the construction needs the number of nodes to be sufficiently large. It is, therefore, often convenient to use a probabilistic construction of an expander.

A natural choice for an expander graph is a *random $d$-regular graph* on $n$ vertices, formed by taking $d/2$ independent uniform permutations on $\{1, 2, \ldots, n\}$. Friedman (2003) proved a conjecture of Alon, establishing that random regular graphs are *weakly-Ramanujan*.

**Theorem C.2 (Friedman 2003)** *Fix $\epsilon > 0$ and an even integer $d > 0$. Then, suppose $G$ is a random graph generated by taking $d$ independent uniformly random permutations $\pi_1, \pi_2, \ldots, \pi_d$ on $V = \{1, 2, \ldots, n\}$, then choosing the edges as*

$$E = \left\{ (i, \pi_j(i)), (i, \pi_j^{-1}(i)) : 1 \leq j \leq d, 1 \leq i \leq n \right\}.$$

*Then, with probability $1 - O(n^{-\Omega(\sqrt{d})})$, $G$ is a $2d$-regular graph and $\lambda(G) \leq \frac{\sqrt{2d-1}+\epsilon}{d}$.*

In our experiments, we use a random regular graph to instantiate the expander graph component of EXPHORMER. We describe the details below.

**Generating a Random Regular Expander**   Let $G = (V, E)$ be the original graph, where $V = \{1, 2, \ldots, n\}$. Inspired by the expansion properties of the random graph process analyzed in Friedman (2003) (see Theorem C.2), we generate a random regular graph $G' = (V, E')$ on the same node set $V$ as follows.

- Let $s = (1, 1, \ldots, 1, 2, 2, \ldots, 2, \ldots, n, n, \ldots, n)$, where each value appears $d$ times.
- Pick a random permutation $\pi$ on $\{1, 2, \ldots, nd\}$, chosen uniformly at random from $(nd)!$ possible permutations.
- Let $E'$ be the multiset $\{(s_i, s_{\pi(i)}) : 1 \leq i \leq nd\}$.
- Remove any self loops from $E'$; for large $n$, this will happen with probability $o(1)$.
- If $\lambda(G) \leq \frac{\sqrt{2d-1}+\epsilon}{d}$, then stop; otherwise generate a new random permutation $\pi$ and try again.

| Dataset | Model | #Virtual Nodes | | | |
|---------|-------|----|---|---|---|
| | | 1 | 2 | 4 | 8 |
| Cifar10 | Exphormer | $\mathbf{72.33 \pm 0.155}$ | $70.92 \pm 0.176$ | $71.25 \pm 0.219$ | $71.15 \pm 0.19$ |
| | Exphormer+MPNN | $\mathbf{74.75 \pm 0.194}$ | $74.49 \pm 0.177$ | $74.38 \pm 0.145$ | $74.55 \pm 0.22$ |

Table 15: Comparing the effect of the number of the virtual nodes on the final result. Adding more virtual nodes sometimes is beneficial, but also sometimes can make the model overfit and makes it less generalizable.

It is easy to see that this procedure is equivalent to sampling $d$ permutations, so Theorem C.2 shows that the resulting graph will be a $2d$-regular expander with high probability.

## D  UNIVERSALITY OF EXPHORMER

In this section, we detail the universal approximation properties of EXPHORMER-based graph transformers.

One of the limitations of standard message passing networks is that their expressivity is generally confined by the limitations of the WL hierarchy. In other words, they cannot distinguish pairs of non-isomorphic graphs that cannot be distinguished by a suitable WL test.

On the other hand, transformer architectures have the ability to distinguish any graphs.

The work of Yun et al. (2020a) showed that for sequences, transformers are universal approximators, i.e., they can approximate any *permutation equivariant* function mapping one sequence to another arbitrarily closely when provided with enough parameters. A function $f : \mathbb{R}^{d \times n} \to \mathbb{R}^{d \times n}$ is said to be *permutation equivariant* if $f(\mathbf{XP}) = f(\mathbf{X})\mathbf{P}$, i.e., if permuting the columns of an input $\mathbf{X} \in \mathbb{R}^{d \times n}$ results in the columns of $f(\mathbf{X})$ being permuted the same way.

**Theorem D.1 (Yun et al. (2020a))** *Let $1 \leq p < \infty$ and $\epsilon > 0$. For any function $f : \mathbb{R}^{d \times n} \to \mathbb{R}^{d \times n}$ that is permutation equivariant, there exists a transformer network $g$ such that $\ell^p(f, g) < \epsilon$.*

The same work shows an extension to all (not necessarily permutation equivariant) sequence-to-sequence functions that are defined on a compact domain, say, $[0, 1]^{d \times n}$ provided that one uses a positional encoding. More specifically, for any transformer $g : \mathbb{R}^{d \times n} \to \mathbb{R}^{d \times n}$, one can define a *transformer with positional encoding* $g_p : \mathbb{R}^{d \times n} \to \mathbb{R}^{d \times n}$ such that $g_p(\mathbf{X}) = g(\mathbf{X} + \mathbf{E})$. The following results shows that trainable positional encodings allow a transformer to approximate any sequence-to-sequence continuous function on the compact domain.

**Theorem D.2 (Yun et al. (2020a))** *Let $1 \leq p < \infty$ and $\epsilon > 0$. For any continuous function $f : [0, 1]^{d \times n} \to \mathbb{R}^{d \times n}$ that is permutation equivariant, there exists a transformer with positional encoding $g_P$ such that $\ell^p(f, g) < \epsilon$.*

Note that the above theorems hold for *full* (dense) transformers. However, under certain conditions about the sparsity pattern, one can obtain similar universality for sparse attention mechanisms (Yun et al., 2020b).

One important consideration is that the aforementioned results hold for functions on *sequences*. Since we are concerned with functions on *graphs*, it is interesting to ask what the implications are for graph transformers.

We follow the approach of Kreuzer et al. (2021): Given a graph $G$, we can view a node transformer as a function from $\mathbb{R}^{n \times n} \to \mathbb{R}^{n \times n}$ which uses the padded adjacency matrix of $G$ as a positional encoding. Similarly, an edge transformer takes as input a sequence $((i, j), \sigma_{i,j})$ with $i, j \in [n]$ and $i \leq j$ such that $\sigma_{i,j} = 1$ if $i$ and $j$ are connected in $G$ or $\sigma_{i,j} = 0$ otherwise. Any ordering of these vectors corresponds to the same graph. Moreover, we can capture the relevant functions going from $\mathbb{R}^{N(N-1)/2 \times 2} \to \mathbb{R}^{N(N-1)/2 \times 2}$ with permutation equivariance. Ideally, they can be approximated as closely as desired by suitable transformers on the edge input.

Now, simply observe (see Kreuzer et al. (2021)) that one can choose a function (in either the node transformer or edge transformer case) that is (a.) invariant under node permutations and (b.) maps

non-isomorphic graphs to distinct values. We would like to apply one of the above thoerems to such a function.

However, we cannot quite apply Theorem D.1 or Theorem D.2, as it is specifically for full transformers in which all nodes are pairwise connected in the attention interaction graph. The final ingredient we require is a theorem from Zaheer et al. (2020), which gives a universality theorem for sparse transformers on *sequences*.

**Theorem D.3 (Zaheer et al. (2020))** *Let $1 < p < \infty$ and $\epsilon > 0$. For any graph $H$ on $[n]$ that contains the star graph, we have that if $f \in [0,1]^{n \times d} \to \mathbb{R}^{n \times d}$ is a continuous function, then there exists a sparse transformer network $g$ (with trainable positional encodings) such that $\ell^p(f, g) < \epsilon$.*

Now, combining D.3 with the previous observations and noting that EXPHORMER with at least one virtual node contains the star graph, we see that EXPHORMER can *approximate* solutions to the graph isomorphism problem (note, however, that this does not imply a polynomial time solution to the problem, as in Kreuzer et al. (2021)).

