# OpenReview forum: "Exphormer: Scaling Graph Transformers with Expander Graphs"
_ICLR.cc/2023/Conference — Submitted to ICLR 2023_

### Official Review · Reviewer_4gAE · 2022-10-22

**Confidence:** 3
**Correctness:** 3
**Technical Novelty And Significance:** 3
**Empirical Novelty And Significance:** 2
**Recommendation:** 5

**Clarity, Quality, Novelty And Reproducibility:**

Clarity:
The paper is globally well-written with a few caveats:
- It is not clear how the three attention mechanisms are combined.
- Is the model sensitive to the choice of number of virtual nodes and expander degree?

Quality:
- The paper offers interesting mathematical motivations for the use of the expander attention scheme but I don't think the experiments illustrate the benefits of this scheme, which makes the paper somehow imbalanced. It would be more convincing to either study the expander attention pattern and show it leverages the properties detailed in Section 4, or reduce Section 4 to further experimentally study the proposed attention mechanism (sensitivity to hyper-parameters or ablation of local/global/expander attention).

Novelty:
- The use of expander graphs has been recently proposed [1] in the context of GNNs: it would be worth discussing it in the related work to better delineate this contribution. Apart from this, the proposed scheme (combination of local-global-expander attention) seems new.

[1] SPARSIFYING THE UPDATE STEP IN GRAPH NEURAL NETWORKS, Johannes F. Lutzeyer, Changmin Wu, Michalis Vazirgiannis (2021)

Reproducibility: no code is not provided but the appendix seems to contain enough experimental details.

**Strength And Weaknesses:**

Strengths:
- Scaling competitive graph transformers to large graph is an important problem for which this work offers a potential solution.
- This work proposes interesting motivations for the use of expander graphs.
- The proposed mechanism scales to larger graphs (order of magnitude of 100k nodes).
- Some ablations demonstrate the usefulness of Exphormer compared to other graph transformers. More generally, the experimental protocol is thorough (error bars, various baselines and tasks).

Weaknesses:
- Although the use of expander graph is motivated, it is not clear whether these motivations are responsible for the performance of Exphormer: there is to my understanding no ablation on the role of each of the three attention mechanisms (local, global, expander), and the attention pattern could be studied to check whether expander attention indeed uses communication between distant neighbors.
- Experiments on ogbn may be more convincing: experiments on arxiv are behind GAT, and behind GraphGPS for molpcba and code2. Moreover, it would be useful to display the results of Exphormer only on molpcba and code2.



**Summary Of The Paper:**

This work proposes a new attention mechanism for graph transformers aiming at handling graphs with large number of nodes.

More precisely, the attention mechanism is three-fold:
- Local attention, applied to 1-hop neighbors of a given node.
- Global attention via virtual nodes.
- Sparse attention based on an expander graph, which is the main contribution of the paper.
The main model is used within GraphGPS framework, which combines an attention scheme with message passing and positional/structural encodings.

The authors motivate the use of an expander graph based attention by the theoretical properties of expander graphs:
- The expander graph is a good approximation of the complete graph.
- Random walks on the expander graph mix well, meaning that provided there are log(nb_nodes) expander graph attention layers, each node is able to communicate with any other node.

The author then proceed to experimentally demonstrate the benefits of the proposed attention mechanism over various GNNs and Graph Transformers, both in terms of accuracy and number of parameters/scalability. Finally, some ablations are conducted i) by combining other attention schemes than Exphormer with MPNN and ii) by comparing Exphormer without message passing to models that do not necessarily use message passing.

**Summary Of The Review:**

This work offers a sensible solution, a sparse attention scheme relying on expander graphs, to an important problem, scaling graph transformers to large graphs. Expander attention has theoretically suitable properties and the experiments demonstrate that its combination to other attention mechanisms within GraphGPS framework scales to large graphs with reasonable performance in terms of accuracy.

However, it seems to me that the paper stresses on the mathematical motivation of expander attention without studying whether these properties are indeed useful in practice, and the results on ogbn seem to be behind other baselines.

Therefore, I do not recommend acceptance for now. I am willing to increase my score if my concerns (clarification of usefulness of expander attention, discrepancy with GNNs on ogbn-arxiv and with GraphGPS on ogbn-molpcba and ogg-code2) are answered.

---

> ### Author Response · Authors · 2022-11-15
> **Answers to some comments in the review**
>
> Thanks for the comments on ablation studies. We’ve added some to the current appendix; see the general response above for a description.
>
> For ogbn-arXiv results: The leaderboard entries for arxiv are not based on “vanilla” GAT, but also incorporate a number of other techniques, including, mini-batching.  Since the dataset is large, it is hard to fit the entire dataset in memory. As described in the public comment, batching and sampling techniques are orthogonal to our approach; we are proposing a novel architecture, which will benefit further with batching or sampling. We plan to try incorporating these approaches into our model, but believe that is outside the immediate scope of this work.
>
> Anonymized code is now uploaded as well.

---

> > ### Comment · Reviewer_4gAE · 2022-12-09
> > **Acknowledgment**
> >
> > I have read your rebuttal and your comment, thank you very much. My main concern remain: the benefit of expander attention is not clear w.r.t. the motivation described in the paper: it seems on the CIFAR10 ablation that MPNN and local neighborhood do most of the heavy lifting). Therefore, I keep my score as is.

---

### Official Review · Reviewer_X95y · 2022-10-23

**Confidence:** 4
**Correctness:** 2
**Technical Novelty And Significance:** 2
**Empirical Novelty And Significance:** 2
**Recommendation:** 3

**Clarity, Quality, Novelty And Reproducibility:**

Clarity: This work is well-organized, well-written, and the figures and the diagrams are helpful to understand the model architecture.

Quality: The theoretical analysis of expander graphs is solid and complete. However, the motivation of using graph transformers and expander graphs is insufficient, and the lack of large-scale dataset experiments is not convincing.

Novelty: This work combines graph transformers and expander graphs, which is innovative.

Reproducibility：The authors did not provide supplementary material, so it cannot be reproduced.



**Strength And Weaknesses:**

Strengths:
1. Nice English writing and clear sentences.
2. The figure provides a good overview of the method.
3. The theoretical analysis of expander graphs is solid and complete.
4. Compared to other graph transformers, Exphormer has significant advantages in computational time complexity.

Weaknesses:
1. Insufficient motivation of using graph transformers as encoders. In the formula of the generalized (dot-product) attention mechanism at the bottom of page 3, each node only calculates attention scoers with its own 1-hop neighbor nodes. Such an attention process is not much different from the graph attention network (GAT), and destroys the properties of the global receptive field of graph transformers, which is a key advantage of graph transformers towards GNNs. There seems no problem if the graph transformer here is replaced with GAT, so the motivation of using graph transformers as encoders is not sufficient.
2. The authors discuss in section 4.2.2 that if a certain number of transformer layers are stacked, the global receptive field of the Exphormer can be recovered. However, this method is common in GNN. Besides, the authors mention in the introduction that the stacking can lead to problems such as oversmoothing and oversquashing. It is mentioned above that the graph transformer in Exphormer is not very different from GAT, so will Exphormer also face the problems of oversmoothing and over squashing if stacking multiple layers?
3. The authors mention in the introduction that most of the experiments done by GraphGPS focused on small graphs, but this work only conducts experiments on one large dataset, ogbn-arxiv. A key advantage of Exphormer is scalability, so it needs to be supplemented with more node classification experiments on other large-scale graph datasets.
4. Theorem 4.1 analyzes the  approximation properties of expander graphs to complete graphs, but complete graphs are only a kind of special structures, which cannot represent the approximation propertities for general graph structures, and it seems there is no proof for this theorem. Besides, $S$ in equation 1 has no annotation, $\pi^{(t+1)}=D_{G}^{-1}A_{G}\pi^{(i)}$ in section 4.2.2 should be replaced with $\pi^{(t+1)}=D_{G}^{-1}A_{G}\pi^{(t)}$.
5. In Appendix D, the authors analyze the universal approximation properties of Exphormer. However, the authors only uses the global attention module. It seems that expander graph attention and local neighborhood attention modules do not contribute to the theoretical performance of the model.


**Summary Of The Paper:**

This work proposes Exphormer, a framework to improve the performance and scalability of graph transformers. Exphormer consists of three components, including expander graph attention, local neighborhood attention and global attention. The key innovation of this work is that Exphormer introduces expander graphs on graph transformers as a graph data augmentation method, which has rich theoretical properties. Meanwhile, Exphormer uses a sparse attention technology similar to BigBird, which reduces the quadratic time complexity of self-attention, making it possible to use it on larger-scale data. The authors demonstrate the abality and efficiency of Exphormer from two aspects of theoretical and experimental analysis, and achieves sota results on three datasets.

**Summary Of The Review:**

This work combines graph transformers and expander graphs together, and uses sparse attention mechanism to reduce quadratic time complexity. However, the motivation of using graph transformers and expander graphs is insufficient, and the lack of large-scale dataset experiments is not convincing.

---

> ### Author Response · Authors · 2022-11-15
> **Responses to comments in the review.**
>
> Thank you for taking the time to provide valuable comments.
>
> > 1. Insufficient motivation of using graph transformers as encoders [...] Such an attention process is not much different from the graph attention network (GAT)
>
> The graph transformer architecture we propose is quite different from GAT, which is a message-passing GNN that uses attention to place weights on neighbors in the _input graph_. In Exphormer, on the other hand, the k-hop neighbor nodes over which dot product attention is calculated are not from the input graph but from the _attention graph_. The attention graph consists of three components — in addition to local neighborhood edges from the input graph, it also uses (a) expander edges and (b) connections to virtual nodes. Edges in (a) and (b) are used to maintain a global receptive field.
>
> > 2. will Exphormer also face the problems of oversmoothing and over squashing if stacking multiple layers?
>
> Oversmoothing and oversquashing are very interesting theoretical questions, which we certainly don’t have full answers to here – further study probably merits its own paper. If we used only global virtual nodes, or only graph neighbors (depending on the shape of the graph), oversquashing would surely be a serious problem. Part of the intuition for using expander attention edges is that this can help spread information transfer across the graph, and avoid these problems.
>
> > 3. so it needs to be supplemented with more node classification experiments on other large-scale graph datasets.
>
> Thanks for this suggestion; in the global comment above, we mention initial results on datasets from Long Range Graph Benchmarks.
>
> > Theorem 4.1 analyzes the approximation properties of expander graphs to complete graphs, but complete graphs are only a kind of special structures, which cannot represent the approximation propertities for general graph structures, and it seems there is no proof for this theorem. Besides, $S$ in equation 1 has no annotation, $\pi^{(t+1)}=D_{G}^{-1}A_{G}\pi^{(i)}$ in section 4.2.2 should be replaced with $\pi^{(t+1)}=D_{G}^{-1}A_{G}\pi^{(t)}$.
>
> Indeed, there was a typo in the indexing, and we have corrected the formula for $\pi^{(t+1)}$. Regarding the approximation properties, the motivation is as follows: In order to make use of a wide global receptive field, graph transformer architectures typically use full connectivity for the dot-product attention mechanism, i.e., information propagation is over the complete graph. Since a sparse attention mechanism will inherently remove edges from the complete graph, this motivates whether the sparse graph can approximate spectral properties of the complete graph.

---

> > ### Comment · Reviewer_X95y · 2022-12-06
> > **Response to author feedback**
> >
> > Thanks for the author's reply. Considering most concerns still remain, I would maintain my original ratings

---

### Official Review · Reviewer_U7gQ · 2022-10-24

**Confidence:** 4
**Correctness:** 3
**Technical Novelty And Significance:** 2
**Empirical Novelty And Significance:** 2
**Recommendation:** 3

**Clarity, Quality, Novelty And Reproducibility:**

Clarity: The writing is not very clear and not friendly for readers.
Quality: The related theoretical analysis maybe reasonable, but the experimental results cannot validate the efficacy of the proposed method.
Novelty: A transferable work from BigBird, and the effectiveness remains uncertain.
Reproducibility: Not Applicable.


**Strength And Weaknesses:**

Strengths:
-	Comparing with full-attention graph Transformers, Exphomer can perform sparse attention mechanism on larger graphs and efficiently reduce the computational cost.
-	 The paper provides plenty of theoretical analysis to support their claims, although I have some concerns.
Weaknesses:
-	For two graphs with the same number of nodes, using the same degree constant may generate the same regular graphs, thus preserving the same edges.  How does this help in learning the features of the two graphs? In fact, the paper doesn’t distinctly illustrate the function of expander graph attention for learning node representations.
-	The degree of regular graphs, d, is an important hyperparameter and graphs of different sizes may need different d, especially on large graphs. However, the parameter studies of d on different graphs are missing.
-	Lacking of ablation studies to verify the effectiveness of the three main components of Exphormer. I’m curious about how much performance gain can expander graph attention bring.
-	More experiments need to be conducted on large graph benchmarks (Citeseer, Pubmed, Computer, ogbn-proteins, etc.) to validate the effectiveness and scalability (especially the comparison of time and memory cost) of the proposed method on large graphs. In addition, existing sparse attention methods (GT[1]) and sampling-based methods (NAGphormer[2], Gophormer[3]) shall be chosen as baselines on large graphs.
-	The paper is not very easy to follow and some notations are confusing. And the formulas are not numbered. Specifically, the format of Table 1 is totally a chaos and need to be reorganized.

[1] Dwivedi V P, Bresson X. A generalization of transformer networks to graphs[J]. arXiv preprint arXiv:2012.09699, 2020.
[2] Chen J, Gao K, Li G, et al. NAGphormer: Neighborhood Aggregation Graph Transformer for Node Classification in Large Graphs[J]. arXiv preprint arXiv:2206.04910, 2022.
[3] Zhao J, Li C, Wen Q, et al. Gophormer: Ego-Graph Transformer for Node Classification[J]. arXiv preprint arXiv:2110.13094, 2021.


**Summary Of The Paper:**

This paper presents a new and scalable graph Transformer, named Exphormer. The motivation of this work is that existing graph Transformers can not efficiently scale to large graphs. Exphormer designs a sparse attention mechanism that consists of three main components: expander graph attention, local neighborhood attention and global attention. As the main building block, expander graph attention utilizes the edges of a regular expander graph with a constant degree in the interaction graph, thus allowing it have a linear computational cost. The authors further provide theoretical analysis of expander graphs and the implication with Exphormer. Extensive experimental results demonstrate the effectiveness of the proposed Exphormer.

**Summary Of The Review:**

The method and theoretical analysis look good, but the experimental analysis, comparisons and writings need to be improved.

---

> ### Author Response · Authors · 2022-11-15
> **answers to the questions raised in the review.**
>
> Thank you for taking the time to provide us with insightful comments. We address your concerns below:
>
> > Q: For two graphs with the same number of nodes, using the same degree constant may generate the same regular graphs, thus preserving the same edges. How does this help in learning the features of the two graphs? In fact, the paper doesn’t distinctly illustrate the function of expander graph attention for learning node representations.
>
> We note that while two different graphs with the same number of nodes may indeed use the same expander graph, this does not generally lead to identical dynamics for both graphs. Expander edges are just one component of Exphormer, which also makes use of _local neighborhood edges_ that vary between the two input graphs. Additionally, node and edge features vary between the graphs. Attention edges primarily determine how information is propagated and, in fact, in standard Graph Transformer architectures that use _full attention_ (e.g., the full transformer in GraphGPS), the attention edges are identical for all graphs, regardless of size.
>
> > Q: The degree of regular graphs, d, is an important hyperparameter and graphs of different sizes may need different d, especially on large graphs. However, the parameter studies of d on different graphs are missing. - Lacking of ablation studies to verify the effectiveness of the three main components of Exphormer. I’m curious about how much performance gain can expander graph attention bring.
>
> Please see answers in our general response above.
>
> > Q: ​​The paper is not very easy to follow and some notations are confusing. And the formulas are not numbered. Specifically, the format of Table 1 is totally a chaos and need to be reorganized.
>
> We note that the format of Table 1 largely follows that of GraphGPS. Could you please clarify why you find the table chaotic and how its readability could be enhanced? We’d be happy to restructure the table to make it more readable if you have any suggestions for how to do so. Moreover, feedback on specific notation that you found confusing would greatly help us improve the readability..
>
> > Q: Quality: The related theoretical analysis maybe reasonable, but the experimental results cannot validate the efficacy of the proposed method. Novelty: A transferable work from BigBird, and the effectiveness remains uncertain.
>
> As described earlier in the public comment, a vanilla implementation of BigBird performs poorly on graph datasets for a few reasons: particularly, it has no clear definition of neighborhood, and ends up with too many parameters. GraphGPS uses a standard implementation of BigBird, which performs uniformly worse than Exphormer on all datasets. Our application of expander graphs performs better than BigBird (and also Performer) in practice, and provides stronger theoretical guarantees.
> Please also see the general response above for comments on interpreting the performance results. As mentioned there, we’ve also added new ablation studies to help pick out which parts of Exphormer help achieve improved performance.

---

> > ### Comment · Reviewer_U7gQ · 2022-12-13
> > **Response to author feedback**
> >
> > Thanks for the author's reply. Considering most concerns still remain, I would maintain my original ratings.

---

### Official Review · Reviewer_eDs8 · 2022-10-24

**Confidence:** 4
**Correctness:** 3
**Technical Novelty And Significance:** 3
**Empirical Novelty And Significance:** 2
**Recommendation:** 6

**Clarity, Quality, Novelty And Reproducibility:**

__Clarity__:
1. What is the difference between the BigBird model used in GraphGPS and Exphormer? Do they only differ in the choice of the random expander graphs?
2. Exphormer seems to work well mostly on synthetic and image datasets. On molecular graph datasets (ogbg-molpcba), there is a clear performance gap between Exphormer and GraphGPS, could the authors elaborate more on why Exphormer does not work well for molecular graphs? Also, why there is no standard deviation available in Table 5 for Exphormer?
3. What are the search grids for different hyperparameters?
4. Is Exphormer permutation-invariant? At inference time, does Exphormer still use a random expander graph? If so, how to ensure generating the same output for the same graph with different permutations of nodes.

__Quality__:
1. The work addresses an important and challenging problem for graph transformers, showing promising empirical results.
2. The implementation descriptions are detailed, and the experimental setup is fair and sound.

__Novelty__:
- The use of d-regular expanders is original. However, it would be important to clarify the difference between the BigBird and Exphormer in experiments.

__Reproducibility__:
- All hyperparameters are provided, but the code was not provided.

**Strength And Weaknesses:**

##### Strengths
1. The idea of using expander graphs (specifically d-regular expanders for fixed d) as a sparse approximation of complete graphs is elegant and original to my best knowledge.
2. The proposed method seems to be simple to implement and to incorporate into existing framework combining transformers and GNNs, which only requires a sampler for generating random regular expander.
3. The empirical comparison is fair and solid.
4. The paper is well-written and easy to follow.

##### Weaknesses
1. (Positioning) This work is motivated by the fact that most existing graph transformers are limited to small graphs due to the quadratic complexity of full dot-product attention. However, the conducted experiments are somewhat misleading as most of the results are only shown on datasets of small-scale graphs. Given marginal improvements over the previous SOTA transformer GraphGPS that used either full or linear attention (e.g. Performer), the impact of this method is not very clear. On the other hand, for the more interesting tasks with large-scale graphs like ogbn-arxiv, the paper only shows very preliminary results, far behind the SOTA results. I expect a thorough evaluation of Exphormer on more datasets with large-scale graphs, such as other node classification datasets of OGB.
2. (Limited task types) Exphormer is only evaluated on classification tasks. It would be more convincing to demonstrate the effectiveness of Exphormer on graph regression tasks.
3. (Missing experiments) Some important studies of hyperparameters are missing, including the effect of expander graph degrees, the effect of the number of virtual nodes. It would also be useful to show the memory usage of the vanilla transformer, BigBird, Performer, and Exphormer.


**Summary Of The Paper:**

The paper proposes a new sparse attention mechanism for graph-structured data to address the quadratic complexity issue of graph transformers. Inspired by the idea of the sparse attention module BigBird for sequences, the proposed method, named Exphormer, also constructs an interaction graph based on three types of edges for attention. These edges include a random expander graph (random attention), the original graph (local neighborhood attention), and edges between any node and a virtual node (global attention). Thanks to the fruitful mathematical properties of expander graphs, the authors prove mixing properties and a universal approximation property of Exphormer. The authors show the first time that BigBird-like models, namely Exphormer, achieve very good performance on five graph benchmark datasets.

**Summary Of The Review:**

This work successfully extends BigBird to graph-structured data and demonstrates the first time that BigBird-like sparse attention achieves SOTA performance on several graph classification datasets. I have some concerns about its insufficient experimental results on large-scale graphs, limited task types considered, and some missing experiments relative to hyperparameters and memory usage.

Overall, I think the strong points overweight the weak points and I recommend a weak accept. I would further increase my score if all my concerns and questions are answered.

---

> ### Author Response · Authors · 2022-11-15
> **responses to specific review comments**
>
> Thank you for the review and the helpful comments.  Here are some responses to some of the comments in the review.
>
> ## For the comments in the weaknesses:
>
> 1. Positioning
> Please see the first general comment above about how our work relates to the other models in this area.
>
> 2. Limited task types: Please see a discussion on the datasets in the public comment. There are very few regression tasks in standard graph datasets. The most widely used one is ZINC, which we avoided for reasons discussed in the public comment above.
>
> 3. Thanks for bringing this to our attention. We have now added an additional table with hyperparameters for the additional datasets and have also provided training time and memory usage information. Furthermore, we are adding the results of ablation studies on the various components (e.g., expander degree, virtual nodes) of Exphormer.
>
> ## For the “clarity” comments:
>
> 1. Please see the general response above for comparison to BigBird.
>
> 2. Molecular graphs tend to be on the small side, e.g., ogbg-molpcba has an average node size of just 26. On small graphs, there isn’t much of a difference in the full attention mechanism and the attention mechanism of Exphormer (and, in fact, for extremely small graphs where the number of nodes is less than d, Exphormer defaults to the full attention mechanism by default). We nevertheless reiterate that even in the case of datasets over smaller graphs, a fairer comparison would be between Exphormer and linear attention models (e.g., BigBird and Performer).
> Thank you for pointing out the missing standard deviations. We have updated the draft with this information.
>
> 3. Due to the large number of hyperparameters, as in GraphGPS, we cannot perform a full grid search. Our search was mostly based on linearly searching over some parameters and making educated guesses on others, based on the hyperparameters from GraphGPS or looking into the learning curves.
>
> 4. Exphormer is permutation-invariant in the sense that the distribution of possible attention graphs is the same, no matter what order the graph is shown in. Given two permutations of the same graph, however, fixing the random seed or similar processes won’t ensure the same particular realization of the random graph is selected.
>
>
> ## Comments in the Reproducibility section
> Exphormer was developed as a Github project by forking GraphGPS code. Our implementation will be made public after the review process is complete for anonymity reasons. We will upload the code (without author identifications) as supplementary material for the reviewers to inspect the code.

---

> > ### Comment · Reviewer_eDs8 · 2022-12-12
> > **Response**
> >
> > Thank you for the response. While my initial score was weak acceptance, I agree with other reviewers that several concerns still remain, such as non-convincing results on ogbn-datasets, limited task types, and missing theoretical understanding of the proposed method. I think the paper could be much strengthened if it can address these isssues.

---

### Author Response · Authors · 2022-11-15
**Comparing results with other papers**

Thanks to the reviewers for the comments. We will emphasize some of these points in the main paper in future drafts. Here are some comments relevant to all reviewers; we’ll also respond to individual reviewers separately.

## Comparing results with other papers

First, a note on comparing the results of our paper with those of other papers/models. Exphormer has been designed to scale graph transformers to much larger datasets.

### Larger datasets
Previous Transformer-based models like SAN and Graphormer ran their experiments only on smaller graphs (~100 nodes per graph).  GraphGPS took the first steps towards improving scalability by combining a transformer (either full attention, or linear variants Performer and BigBird) with a message-passing network (MPNN). They ran their experiments on some larger datasets (particularly MalnetTiny, whose graphs have about 1500 nodes on average, 5000 nodes maximum).

However, GraphGPS’s performance using linear transformers is relatively poor. Their best results are obtained by using only the full transformer (see their Appendices A and B).  Moreover, their ablation studies show that a significant part of their improvement comes from the MPNN component (CustomGatedGCN); this is actually a rather large model, with more parameters than the transformer component.

We show that Exphormer model performs better than the linear models (BigBird and Performer), Tables 2, 9-12.

Compared to GraphGPS, we do in fact show some nontrivial improvements.

1. Parameters and memory usage
We use less memory than GraphGPS, even their linear transformer versions. For instance, on Malnet-tiny, GraphGPS with a full transformer only runs with a batch size of 8 on a 40GB A100 GPU. Meanwhile, our model easily fits with a batch size of 16 into a 32GB V100 (using only 60% of memory). We will add memory usage comparisons in the appendix in the next draft of the paper.

2. Accuracy
Most of GraphGPS’s best results are based on full transformers, not linear variants (see tables A3-5 in the GraphGPS paper). In some cases, we do beat full-attention GraphGPS models. But, for fair comparison with Exphormer+MPNN in terms of memory usage and computation time, it’s better to only compare the linear models, where Exphormer models have fewer parameters, run faster, and give far better results than GraphGPS models based on BigBird or Performer.

### Comparison to other models
NagPhormer achieves scaling using other techniques like batching and sampling, which are orthogonal to our approach: our conceptually novel approach to scaling transformers could also incorporate batching and sampling techniques to scale to even larger datasets. Since Arxiv is a large dataset, we are constrained by memory on the GPU, and show results for models that fit in memory. Adding batching and sampling could improve performance, but we consider that somewhat outside the scope of our improvements to attention mechanisms.

---

> ### Author Response · Authors · 2022-11-15
> **Choice of datasets**
>
> Since we want to demonstrate scalability, we focused on larger datasets. In GraphGPS, especially on small molecular datasets such as ZINC, the transformer part not only does not improve the results, but sometimes just introduces unnecessary complexity, which leads to worse results on the test dataset. Thus, we have mostly avoided these datasets.
>
> Based feedback here and the recent addition of Long Range Graph Benchmark datasets (LRGB) to the GraphGPS paper (12th Oct version of https://arxiv.org/abs/2206.08164), we’re now adding experiments on graphs with larger diameter. Doing full experiments and finding good hyperparameters will take some time, but so far, we’ve obtained 37.88 ± 0.392 for VOCPascal-SP, much better than the SOTA by GraphGPS. For a molecular-type dataset with a regression task, we’ve experimented with peptide-struct dataset; our best initial result is MAE of 0.257, comparable to GraphGPS’s best model. Due to the large space of parameters, a full grid search is impractical, and so the process of manually refining hyperparameters to achieve SOTA takes some time. We will report progress by the end of the rebuttal phase.

---

> > ### Author Response · Authors · 2022-11-15
> > **Comparison to BigBird**
> >
> > The BigBird model used in GraphGPS is virtually identical to the one used in sequential tasks, except that (a) one uses an arbitrary sequential order of the nodes (tokens), and (b) one uses a wider array of positional encodings. The arbitrary nature of the node ordering (and the resulting “window attention” component in BigBird) leads to worse performance, compared to the graph-centric sparse attention model of Exphormer.
> >
> > Regarding differences between BigBird and Exphormer, we note that (a) Exphormer replaces the “window attention” of BigBird and (b) uses expander graphs instead of Erdős–Rényi graphs. As noted in the paper, while Erdős–Rényi graphs do sometimes have expansion properties, they typically need a superlinear number of edges to do so. In particular, the instantiation of parameters in BigBird (needed to guarantee a linear attention mechanism) does not even guarantee connectivity of the random graph component; by contrast, the expander graphs used in Exphormer achieve expansion properties with a linear number of edges.

---

> > > ### Author Response · Authors · 2022-11-15
> > > **Ablation studies**
> > >
> > > Thanks to the reviewers who suggested further ablation studies. In our updated draft, we have begun adding a new set of ablation studies, including the effect of the expander graph degree and the number of virtual nodes. We also have listed experiments in which we remove each of the individual attention components, one at a time, to determine the effect of each part of Exphormer. We have added these studies for CIFAR10 in the current appendix, and are continuing to add more studies of this form in the final version.

---

### Decision · Program_Chairs · 2023-01-20

**Decision:**

Reject

**Justification For Why Not Higher Score:**

I think this is a nice paper, but the theoretical motivation is lacking and the reviewers were overall quite lukeworm

**Justification For Why Not Lower Score:**

N/A

**Metareview: Summary, Strengths And Weaknesses:**

The paper proposes a new transformer-type GNN architecture based on expander graphs. This is an interesting and novel idea (expander graphs have made an appearance in several recent GNN works, but overall are rather new in this community), which was also appreciated by the reviewers. However, the reviewers raised concerns about non-convincing results on ogbn-datasets, limited task types, and missing theoretical understanding of the proposed method (although the use of expander graphs is motivated, it is not clear whether these motivations are responsible for the performance of Exphormer). This opinion also prevailed after the rebuttal.
Our recommendation is rejection.